# Hepatic Reirradiation for Patients with Recurrent Hepatocellular Carcinoma

Yaoru Huang [1,2,*], Po-Yung Chen [3,†], Tzu-Yen Cheng [3,†] and Jeng-Fong Chiou [1,4,5,*]

1  Department of Radiation Oncology, Taipei Medical University Hospital, Taipei 110, Taiwan
2  Graduate Institute of Biomedical Materials and Tissue Engineering, College of Biomedical Engineering, Taipei Medical University, Taipei 110, Taiwan
3  School of Medicine, College of Medicine, Taipei Medical University, Taipei 110, Taiwan; b101104056@tmu.edu.tw (P.-Y.C.); b101104073@tmu.edu.tw (T.-Y.C.)
4  Department of Radiology, School of Medicine, College of Medicine, Taipei Medical University, Taipei 110, Taiwan
5  Taipei Cancer Center, Taipei Medical University, Taipei 110, Taiwan
*  Correspondence: 967092@h.tmu.edu.tw (Y.H.); solomanc@tmu.edu.tw (J.-F.C.)
†  These authors contributed equally.

**Featured Application: Hepatocellular carcinoma (HCC) may easily recur because of multifocal carcinogenesis. For recurrent HCC, good local control indicates better survival. For HCC-contradicting local therapies such as surgery, transarterial chemoembolization and radiofrequency ablation, several studies have discussed the feasibility of reirradiation. This study investigated the treatment outcomes of reirradiation and discovered better local control may still prolong survival after reirradiation for recurrent HCC. This may suggest a new treatment approach for treating HCC.**

**Abstract:** For treating hepatocellular carcinoma (HCC), local therapies and surgery, including liver transplant, are the first line treatment options; however, several contraindications limit their clinical use. The improvement of radiotherapy (RT) established RT in treating HCC contraindicated against local therapies, including transarterial chemoembolization and radiofrequency ablation. For HCC that recurs after RT and still contradicts against local therapies, there is a need to investigate the use of reirradiation. This study recruited patients receiving two courses of RT for recurrent HCC between January 2007 and December 2019. The result suggested that patients who experienced tumor regression after reirradiation had better survival over those with a stable form of the disease, with the mean overall survival (OS) as 30.0 and 4.0 months, respectively ($p < 0.001$). The analysis also revealed that systemic therapy had no benefit on both the OS and controlling distant metastasis; the result was limited to a small study number and diversity of drugs. Considering systemic therapy and portal vein tumor thrombosis, which are commonly viewed to affect prognosis, multivariate analysis suggested that the Child–Pugh score and local control were the only two independent factors for the OS, with $p = 0.017$ and $p = 0.028$, respectively. Our findings suggested that reirradiation could be the choice for treating recurrent HCC.

**Keywords:** hepatocellular carcinoma; radiotherapy; reirradiation; survival

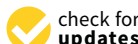



## 1. Introduction

Hepatocellular carcinoma (HCC) is one of the most common and lethal malignancies over the world [1,2]. Hepatitis B or C-induced liver cirrhosis, the most frequent risk factor for HCC, causes multifocal carcinogenesis and consequently results in tumors developing in different hepatic lobes synchronously or metachronously [2,3]. Local therapies, including surgical excision, transarterial chemoembolization (TACE), radiofrequency ablation (RFA) and cryotherapy, may temporarily control the tumor [1,3,4]; however, larger tumors with

little liver reserve, tumor numbers, tumor location and portal vein tumor thrombosis (PVTT) are common contraindications against the local therapies mentioned above. A liver transplant is the solution for certain conditions as so; however, the difficulty of the surgery and the availability of liver donation limit the clinical use [4].

Recent improvements in radiotherapy (RT) have established RT in treating HCC, especially for larger tumors, tumors close to the diaphragm or in the presence of PVTT [4]. Advancing RT techniques, from traditional three-dimensional conformal radiotherapy (3D-CRT) to intensity-modulated radiotherapy (IMRT), along with use of image-guided radiotherapy (IGRT), respiratory motion management and even stereotactic ablative radiotherapy (SABR), make it feasible to deliver higher radiation dose to the target and constrain lower doses to the normal liver tissue. The progress results in better tumor control with fewer toxicities [1,5]. For HCC that recurs after RT and is still contraindicated against other local therapies, several studies have discussed the feasibility of repeated irradiation to the liver. One Korean study retrospectively presented the feasibility of using reirradiation to treat recurrent HCC in a 45 patients with the traditional 3D-CRT technique, while another Taiwanese study categorized liver function as a predicting factor for radiation-induced liver disease (RILD) after hepatic reirradiation for HCC [6,7]. Massachusetts General Hospital (MGH) also demonstrated a safe reirradiation to either primary or secondary hepatic malignancies with only 4.1% of patients developing RILD [8]. In 2020, Dr. Owen concluded a treatment recommendation for liver reirradiation through a case-based discussion with four radiation oncology experts [9]. Building on the research mentioned above, we investigated the treatment outcome of reirradiation of recurrent HCC; meanwhile, survival and toxicity were also reviewed in this retrospective study.

## 2. Materials and Methods

### 2.1. Patients and Radiotherapy

Patients with a history of HCC who received 2 courses of RT for recurrent, intrahepatic HCC at our institution between January 2007 and December 2019 were included in this retrospective review; the study was approved by the institutional review board. The second course of RT involved either irradiating the same hepatic lobe, different location or a combination of previously treated and new area. Gross tumor volume (GTV) was contoured based on contrast-enhanced computed tomography (CT) images at simulation before RT. Clinical tumor volume (CTV) or internal target volume (ITV) was added according to clinical judgement of diagnostic CT images or magnetic resonance images (MRI) and the use of respiratory motion management, such as fiducial marker or 4-dimentional CT scan. If there was more than one tumor, each would be identified as CTV1 (ITV1), CTV2 (ITV2) and so on; all tumors treated in one RT treatment plan would be viewed as the same RT course. The RT was delivered through IMRT, volumetric modulated arch therapy (VMAT) and tomotherapy, assisted with daily image-guidance. The RT dose was prescribed in conventional or ablative settings, as well as in consideration of dose constraints of the normal organs: conventional dose was 50 Gy in 1.8 to 2.5 Gy per fraction (Gy/Fx) or 45 Gy in 3 Gy/Fx while SABR was 8 to 10 Gy/Fx, with total 5 fractions.

### 2.2. Follow-Up and Evaluation of Toxicity

Treatment response was evaluated on triphase CT scan or MRI 4 to 6 weeks after completion of reirradiation. Tumor response was determined under the criteria of the modified Response Evaluation Criteria in Solid Tumors (mRECIST) for HCC. Follow-up image study was scheduled on the basis of 3-month interval. Survival was calculated since completion of reirradiation. Complete blood count and biochemistry profiles including liver function test were checked on the basis of a 4-week visit. Toxicities were assessed according to the Common Terminology Criteria for Adverse Events (CTCAE) ver. 4.0. Classic RILD clinically presents with signs and symptoms including fatigue, abdominal pain, increased abdominal girth, hepatomegaly, anicteric ascites; here it was defined as anicteric hepatomegaly and ascites with elevation of alkaline phosphatase, increasing more

than two times the normal level [7]. Nonclassic RILD manifests as markedly elevated serum transaminases (a more than fivefold increase compared to upper limit of normal) and jaundice with elevation of total serum bilirubin, greater 2.5 mg/dL [7]. Both types of RILD were determined according to lab data and medical records.

### 2.3. Statistics

Overall survival (OS) and progression-free survival (PFS) were calculated by Kaplan–Meier method while Cox regression was for assessing the prognostic factor for survival. Logistic regression and Fisher's exact test were used for evaluating predicting factors for local control (LC). Statistical analysis was performed by SPSS 18 (SPSS Inc., Chicago, IL, USA).

### 3. Results

A total of 33 patients completed two courses of RT for intrahepatic HCC while 32 patients completed image evaluation; the other one patient had no image study because of lethal upper gastrointestinal bleeding. Four of these patients received the third course of RT for newly developed tumor during follow-up; one patient received the third course sequentially with full RT dose to a different tumor at the other hepatic lobe, two weeks after completing the second course. The mean follow-up was 22.4 months, ranging from 1.5 to 127.0 months. The interval between two RT courses was 16.5 months, ranging from 0.5 to 136.0 months (Table 1). No patients died within 1 month after reirradiation while one patient experienced nonclassic RILD with elevated alanine aminotransferase (ALT) and aspartate aminotransferase (AST) over 200 U/L. This patient recovered from RILD and expired due to upper gastrointestinal bleeding, a complication of liver cirrhosis, eight months after the second course of RT. For all patients, the one-year OS was 62.5% and two-year OS was 34.4% (Figure 1A). In total, 30 patients experienced infiled (IF) recurrence, which was defined as reirradiated area; 19 patients had outfield (OF) recurrence, as intrahepatic but nonirradiated area, and 10 patients developed distant metastasis (DM), as extrahepatic lesions (Supplementary Figure S1).

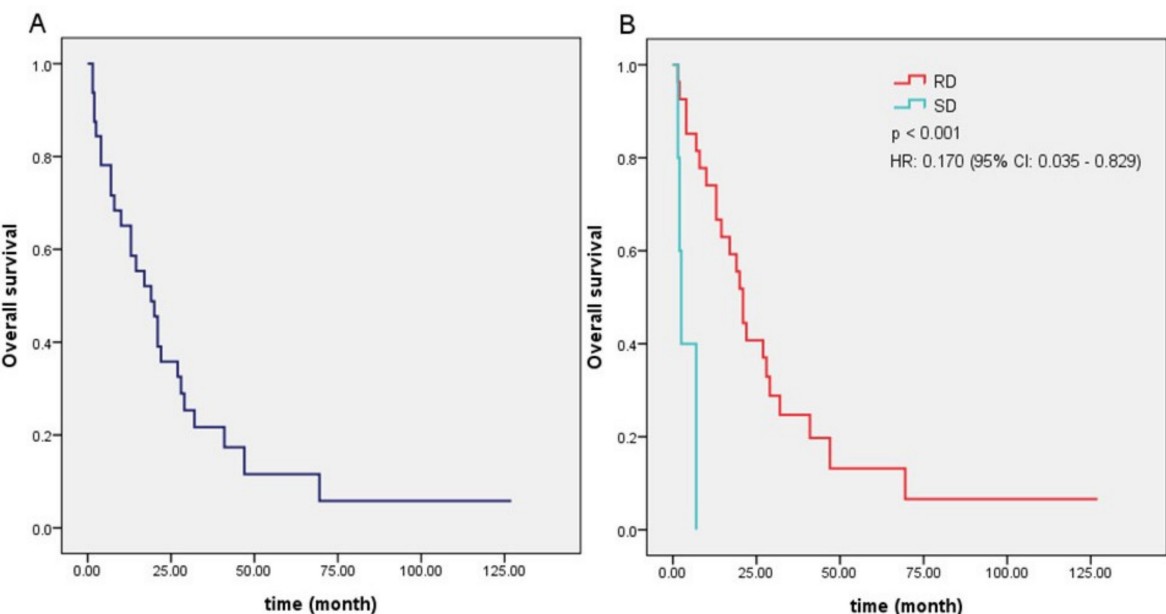

**Figure 1.** Overall survival for (**A**) all patients over two years and (**B**) regression disease (RD) subgroup and stable disease (SD) subgroup.

**Table 1.** Patient characteristics.

| Characteristics | No. of Patients | % |
|---|---|---|
| Patient | 32 | |
| Age (years) | Mean: 66.0, (32–89) | |
| Gender | | |
| 　Male | 29 | 90.6 |
| 　Female | 3 | 9.4 |
| Tumor stage | | |
| AJCC [1] (7th ed., 2010) | | |
| 　T1 | 3 | 9.4 |
| 　T2 | 15 | 46.9 |
| 　T3 | 9 | 28.1 |
| 　T4 | 5 | 15.6 |
| BCLC [2] stage | | |
| 　A | 5 | 15.6 |
| 　B | 15 | 46.9 |
| 　C | 12 | 37.5 |
| Tumor numbers | | |
| 　One | 27 | 84.4 |
| 　Two | 5 | 15.6 |
| Follow-up (months) | Mean: 22.4, (1.5–127.0) | |
| Interval between two RT courses (months) | Mean: 16.5, (0.5–136.0) | |
| Child–Pugh score before the second RT | | |
| 　5 | 22 | 68.8 |
| 　6 | 6 | 18.8 |
| 　7 | 4 | 12.5 |
| Hepatitis status | | |
| 　Hepatitis B | 19 | 59.4 |
| 　Hepatitis C | 5 | 15.6 |
| 　Both hepatitis B and C | 1 | 3.1 |
| Main portal vein tumor thrombosis | | |
| 　Present | 9 | 28.1 |
| 　Absent | 23 | 71.9 |
| Local therapies before RT | | |
| 　Nil | 11 | 34.4 |
| 　Operation | 2 | 6.3 |
| 　TACE | 7 | 21.9 |
| 　RFA | 2 | 6.3 |
| 　Operation and TACE | 6 | 18.8 |
| 　TACE and RFA | 3 | 9.4 |
| 　Operation, TACE and RFA | 1 | 3.1 |
| Radiation dose of the nd RT (BED [3], $\alpha/\beta = 10$) | Mean: 52.3 Gy, (ranges, 12–109.5 Gy) | |
| Use of systemic therapy after the 2nd RT | | |
| 　Yes | 8 | 25.0 |
| 　No | 24 | 75.0 |

Abbreviation: [1] AJCC: American Joint Committee on Cancer, [2] BCLC: Barcelona Clinic Liver Cancer stage, [3] BED: biologically equivalent dose.

For the treatment outcome, 8 patients had complete remission (CR) and 19 patients had partial remission (PR) at the image evaluation; these 27 patients were defined as the regressive disease (RD) subgroup. Two patients were suggestive of progressive disease and the other three had no interval change at the image evaluation; these five patients were categorized as the stable disease (SD) subgroup. Overall, one-year LC was 34.4% while two-year LC was 21.9% (Supplementary Figure S2). However, no predicting factor between RD and SD was discovered (Supplementary Table S3).

Comparing RD and SD subgroups, the mean OS was 30.0 and 4.0 months, respectively, $p < 0.001$ (Figure 1B). Additionally, both the OS and infield progression-free survival (IF-PFS), defined as no recurrence within the reirradiated area, between CR and PR had no difference ($p = 0.860$ and $p = 0.940$) (Figure 2). Besides, Child–Pugh score before reRT was also a key factor to the OS; the median OS for the patients with score 5 and score 6 and 7 were 34.4 and 10.7 months, respectively ($p = 0.005$) (Figure 3). The result confirmed that Child–Pugh classification as a survival-predicting model for the patients with liver cirrhosis. For tumor volumes of less than 100 cm$^3$, it seemed that smaller tumors had marginal benefit on OS ($p = 0.059$) and IF-PFS ($p = 0.075$) (Table 2). However, the presence of PVTT in the main portal vein was shown to have no impact on both the OS and LC after RT ($p = 0.099$ and 0.141, respectively) (Supplementary Table S4).

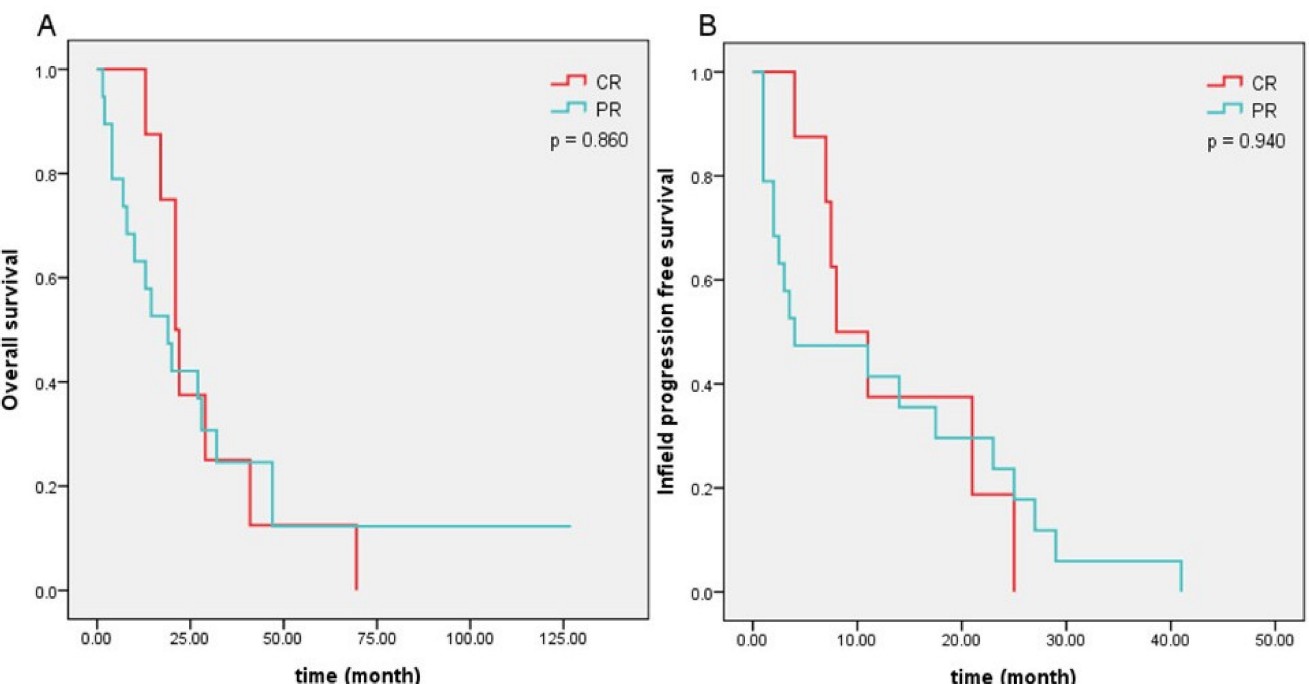

**Figure 2.** (**A**) Overall survival for the patients with complete remission (CR) and the patients with partial remission (PR); (**B**) infield progression-free survival for the patients with CR and the patients with PR.

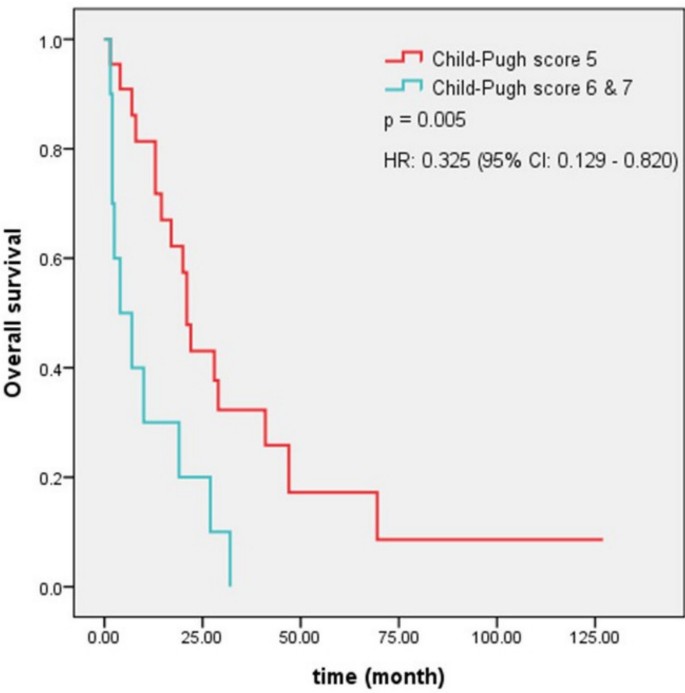

**Figure 3.** Overall survival for Child–Pugh score 5 subgroup and Child–Pugh score 6 and 7 subgroup.

**Table 2.** Univariate and multivariate analysis of infield progression-free survival and overall survival.

| Prognostic Factor | No. of Patient | Infield Progression-Free Survival (IF-PFS) | | | Overall Survival (OS) | | |
|---|---|---|---|---|---|---|---|
| | | Univariate | Multivariate | | Univariate | Multivariate | |
| | *n* (%) | *p* Value | HR [1] (95% CI [2]) | *p* Value | *p* Value | HR (95% CI) | *p* Value |
| Local control | | | | | <0.001 | 0.170 (0.035–0.829) | 0.028 |
| Regression subgroup | 27 (84.4%) | | | | | | |
| Stable subgroup | 5 (15.6%) | | | | | | |
| Regression subgroup | | 0.940 | | | 0.860 | | |
| Complete remission | 8 (25%) | | | | | | |
| Partial remission | 19 (59.4%) | | | | | | |
| BCLC stage | | 0.326 | | | 0.114 | | |
| Stage A | 5 (15.6%) | | | | | | |
| Stage B | 15 (46.9%) | | | | | | |
| Stage C | 12 (37.5%) | | | | | | |
| Use of systemic therapy | | 0.591 | | | 0.513 | | |
| Use | 8 (25.0%) | | | | | | |
| Nonuse | 24 (75.0%) | | | | | | |
| Outfield recurrence and/or distant metastasis | | 0.072 | 3.320 (0.933–11.812) | 0.064 | 0.706 | | |
| Nil | 6 (18.8%) | | | | | | |
| Presence | 26 (81.3%) | | | | | | |
| Child–Pugh score before re-RT | | <0.001 | 0.228 (0.085–0.612) | 0.003 | 0.005 | 0.325 (0.129–0.820) | 0.017 |
| Child–Pugh score 5 | 22 (68.8%) | | | | | | |
| Child–Pugh score 6 and 7 | 10 (31.3%) | | | | | | |
| Tumor volume | | 0.075 | 0.488 (0.172–1.383) | 0.177 | 0.059 | 0.782 (0.299–2.047) | 0.617 |
| Tumor volume $\geq$ 100 (cm$^3$) | 17 (53.1%) | | | | | | |

**Table 2.** *Cont.*

| Prognostic Factor | No. of Patient | Infield Progression-Free Survival (IF-PFS) | | | | Overall Survival (OS) | | |
|---|---|---|---|---|---|---|---|---|
| | | Univariate | Multivariate | | | Univariate | Multivariate | |
| | *n* (%) | *p* Value | HR [1] (95% CI [2]) | *p* Value | | *p* Value | HR (95% CI) | *p* Value |
| Tumor volume < 100 (cm$^3$) | 15 (46.9%) | | | | | | | |
| Dose of radiotherapy | | 0.458 | | | | 0.708 | | |
| BED < 50 (Gy) | 17 (53.1%) | | | | | | | |
| BED ≥ 50 (Gy) | 15 (46.9%) | | | | | | | |
| Main portal vein tumor thrombosis | | 0.141 | | | | 0.099 | | |
| Nil | 23 (71.9%) | | | | | | | |
| Presence | 9 (28.1%) | | | | | | | |

Abbreviation: [1] HR: hazard ratio, [2] CI: confidence interval.

Systemic therapy, including chemotherapy and target therapy, might be added after reirradiation (Supplementary Table S5); however, the use of systemic therapy or not was shown to have no benefit on the OS ($p = 0.513$) and IF-PFS ($p = 0.591$) (Figure 4A,B). It also had no benefit on 6-month, 12-month and 24-month IF-PFS with $p = 0.232$, 0.281, and 0.475, respectively (Supplementary Table S6). This suggested local RT had a more important effect than systemic therapy in LC. OF and DM-PFS were used to calibrate the effect of systemic therapy. The association between the use of systemic therapy with OF and/or DM-PFS was statistically insignificant ($p = 0.423$) (Supplementary Figure S7). Furthermore, progression to OF recurrence and DM had little relation to the OS ($p = 0.706$) (Figure 4C), which might suggest systemic therapy after reirradiation to recurrent HCC had no contribution to survival.

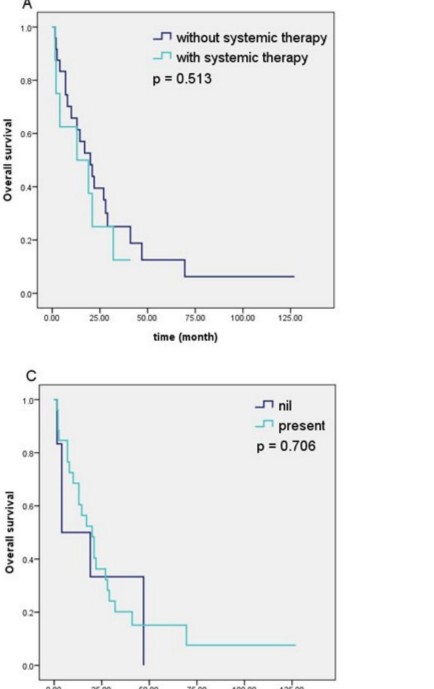

**Figure 4.** (**A**) Overall survival and (**B**) infield progression-free survival for using systemic therapy subgroup and not-using systemic therapy subgroup; (**C**) overall survival for the subgroup with and without outfield recurrence and/or distant metastasis.

Univariate analysis suggested that the use of systemic therapy and the presence of main PVTT had no relation to survival. Despite the tumor stage, only LC and Child–Pugh score were independently associated with the OS under multivariate analysis. RD subgroup could prolong survival over SD subgroup, with hazard ratio (HR) = 0.170, 95% confidence interval (CI): 0.035–0.829, $p$ = 0.028. A Child–Pugh score of 5 suggested better survival than Child–Pugh scores of 6 and 7, with HR = 0.325, 95% CI: 0.129–0.820, $p$ = 0.017. However, tumor stage had little correlation to survival with $p$ = 0.114. This suggested that besides uncompromised liver function indicating longer survival, better LC could also prolong the OS. Furthermore, the Child–Pugh score also had better IF-PFS under multivariate analysis ($p$ = 0.003); treatment outcomes between CR and PR had no statistical difference on OS and IF-PFS (Table 2).

## 4. Discussion

Previous studies had discussed the feasibility of using photon reirradiation to treat recurrent HCCs, but to the best of our knowledge, this was the first study to investigate the treatment outcome considering the use of systemic therapy. Target therapies, including sorafinib, lenvatinib and regrofenib, were proved to have survival benefit after local therapy for unresectable HCC [4]. Systemic therapy, including target therapy, may be added after local RT in real-world practice. Therefore, it is crucial to consider the effect of systemic therapy while evaluating the outcome of reirradiation for recurrent HCC. Two Korean studies did not provide any information about whether they were using systemic therapy or not, while another study from MGH only listed how many patients used it but with no further analysis. Despite the lack of analysis of the use of systemic therapy, 1-yr OS was about 50%, ranging from 57% in Seol's study to 44.9% in McDuff's study [6,8,10]. Moreover, for tumors regressing after reRT, our study revealed a better survival with 1-yr OS up to 74.1% and the mean OS as 30.0 months. For treatment response, Seol shared a total of 62.8% with 18.6% CR and 44.2% PR, using 3D-CRT for reRT, compared with which our total response was 84.4% with 25.0% CR and 59.4% PR [6]. Advances of RT techniques, as well as the use of IGRT and respiratory motion management, are believed to deliver more accurate and precise treatment. The evidence suggests that RT could reactivate the hepatitis B virus, causing fulminant hepatitis with lethal liver failure. For more than half of the patients with chronic hepatitis B, more accurate and precise RT could minimize damage to the normal liver tissue, contributing no lethal liver failure in hepatic reirradiation [11].

Hepatitis, inducing liver cirrhosis, is the major cause of HCC; given that Child–Pugh classification is designed for predicting cirrhosis mortality, our study complied with the principle [7,12]. Similar to Seol el al., whose study suggested Child–Pugh A (score 5 and 6) had better survival than Child–Pugh B (score 7–9) after reirradiation [6,7], our study further analyzed these scores. The result suggested even minorly impaired liver function could still have a negative impact on survival after reirradiation, though Child–Pugh scores 5 and 6 are categorized in the same class A. Moreover, the multivariate analysis suggested LC was an independent contributor to the OS. This is the first study suggesting reirradiation with good LC could prolong survival over previous research. Similarly, repeated RFA for smaller HCC was established to benefit survival rates despite its well-discussed limitations [13,14]. TACE also shared the similar benefits to OS, as well as the same concern regarding its limitations [15]. SABR for HCC was suggested to have better LC over RFA for larger tumor and subphrenic locations [16]. Therefore, repeated RT was eventually proved to contribute to the OS. Additionally, this study discovered that outfield and DM had no effect on OS. Although it might be limited to small recruitment, this finding also corresponded to the studies suggesting better LC contributes to better OS in spite of extrahepatic condition [16].

For subgroup analysis, CR or PR after reirradiation had no difference in either OS or IF-PFS. This may indicate certain limitations of our evaluation. First, we evaluated treatment outcomes through image study 4–6 weeks after treatment, which might not be the best response. From Koong's retrospective study, some tumors may have delayed treatment response and thus, continuous observation for at least 9 months after RT is

recommended [17]. For remarkable PR, it might regress to CR for longer evaluation. Second, MRI with newly developed liver-specific contrast is superior to MRI with the traditional contrast in HCC evaluation, while MRI is already better than CT scan [18]. Most our post-RT evaluation is based on CT scan due to inconvenience of MRI. Third, either CT scan or MRI only reveals anatomic abnormality; however, positron emission tomography (PET) scan reveals more metabolic information about cancer cells. Lu et al. concluded that PET scans have high sensitivity up to 90–100% for detecting recurrent or metastatic hepatic tumors and good evaluation for therapeutic response. Besides, refined PET data may be helpful for avoiding false-positive results after local treatment [19]. Limited by reimbursement, we seldom used PET scans for pre- and posttreatment evaluation. Moreover, there might still be a difference between remarkable and slight PR; objective and effective criteria would be needed for more rigorous evaluation. All these limitations and the nature of the retrospective review deterred a rigorous assessment of treatment outcomes.

Interestingly, the use of systemic therapy was found to have no association to OS; it even had no control benefit on OF and DM-PFS. Besides small study numbers and an even smaller proportion using systemic therapy restricted the analysis, diversity of systemic therapy is hard to correlate with outcome. Target therapies, such as sorafenib, lenvatinib and regrofenib, have recently established survival benefit for unresectable HCC even after local therapy [20–22]; however, the medication was not available for patients in the earlier period, and some patients experienced tumor recurrence after target therapy. Furthermore, some of systemic therapies were shown to have tumor control in phase 2 or smaller, nonrandomized phase 3 studies [23,24]. Some patients would continue to receive these medications under salvage settings. This also complicated the analysis; therefore, we could only take whether patients were using systemic therapy or not as an analytic factor and adjust the effect by observing PFS of DM and OF. Consequently, there is a need to further clarify the effect of systemic therapy under a more exclusive recruitment in further research.

This study was subject to inherent nature of a retrospective study design. Not only the small patient number in an over ten-year period but also the complexity of treatment course limited the analysis. Additionally, toxicity assessment was also expected to be more thorough under a prospective setting. Last, growing evidence has suggested survival benefits of a combination of RT and other local therapies, as well as target therapy [25,26]. Although this study concluded that reirradiation for recurrent HCC may have better LC with survival benefit, well-designed prospective research is needed in future for further investigation on improving treatment strategies for unresectable HCC.

**Supplementary Materials:** The following are available online at https://www.mdpi.com/2076-341 7/11/4/1598/s1, Figure S1: Pattern of disease progression, Figure S2: Local control of all patients, Table S3: Correlation between risk factors and local control; Figure S4: (A). Overall survival for the patients with and without portal vein tumor thrombosis (B). In-field progression-free survival for the patients with and without portal vein tumor thrombosis; Table S5: Systemic therapy after re-irradiation; Table S6: Relationship between in-field progression-free survival and systemic therapy; Figure S7: Out-filed recurrence and/or distant metastasis free survival for subgroup with and without systemic therapy.

**Author Contributions:** Y.H. coconceived the study, participated in statistical analysis and drafted the manuscript. P.-Y.C. and T.-Y.C. contributed equally in retrieving clinical data and patients' info, performing statistical analysis and drafting the manuscript. J.-F.C. conceived and coordinated the study. All authors have read and agreed to the published version of the manuscript.

**Funding:** This research received no funding.

**Institutional Review Board Statement:** The study was conducted according to the guidelines of the Declaration of Helsinki, and approved by the Institutional Review Board of TAIPEI MEDICAL UNIVERSITY JOINT REVIEW BOARD (protocol code N202008017 and date of approval 2020/10/13).

**Informed Consent Statement:** Patient consent was waived due to retrospective study with deidentified analysis.

**Conflicts of Interest:** The authors declare no conflict of interest.

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
