# Peer review of "Hepatic Reirradiation for Patients with Recurrent Hepatocellular Carcinoma"

_applsci, doi:10.3390/app11041598_

Round 1
Reviewer 1 Report
Manuscript review entitled “Hepatic re-irradiation for patients with recurrent hepatocellular carcinoma”
The authors reported the usefulness of 2nd and 3rd RT for recurrent HCCs and the prognostic factors after treatment; however, there are several uncertain points in this manuscript.
General comments
- The details of the cohort is unclear. The authors should clearly describe the patient-tumor backgrounds in this cohort. How many patients did undergo 2nd or 3rd RT for a locally progressed tumor after RT? The diameter of each tumor is also lacking in Table 1.
- I speculate that tumors under several different conditions might be included in this cohort, for example, locally progressed tumors after RT, locally progressed tumors after other treatments (TACE, RFA, and resection), and newly developed tumors at the different site of the liver after RT. The therapeutic effects on abovementioned tumors might be different. The readers want to know whether 2nd RT is effective for the locally progressed tumor after RT. Please describe the details of the efficacy of RT for tumors with different treatment histories.
- I speculate that the same patient might be counted as two different patients in Table 1.
- Discussion seems to be very confusing. Please focus on the efficacy and safety of repeated RT. In addition, the overall OS of this study should be compared with other treatments, such as systemic therapy, radioembolization, hepatic infusion chemotherapy, and TACE.
Abstract
- “For treating hepatocellular carcinoma (HCC), liver transplant is the top treatment choice but difficulty of the surgery and availability of liver donation limit the clinical use.”
I do not think so. Surgical resection and ablative therapy are also curative treatment. Please revise the sentence.
- “Considering systemic therapy and portal vein tumor thrombosis, multivariate analysis only suggested Child-Pugh score and local control were the only two independent factors for the OS with p=0.036 and p=0.022, respectively.”
Why are the words “considering systemic therapy and portal vein tumor thrombus” necessary?
Materials and Methods
- “The RT was delivered through IMRT, volumetric modulated arch therapy (VMAT) and tomotherapy, assisted with daily image-guidance.”
The cohort included the patients who were treated with RT in January 2007. Were some early cases treated with conventional RT, not IMRT? Please clarify.
- “Treatment response was evaluated on tri-phase CT scan or MRI 4 to 6 weeks after completion of re-irradiation. Tumor response was determined under the criteria of the modified Response Evaluation Criteria in Solid Tumors (mRECIST) for HCC”
In my experience, tumor vascularity usually remains for a relatively long-term after RT. I think that it is very difficult to judge the local tumor response on CT or MRI obtained 4-6 weeks after RT.
- Please spell out “RILD”.
Results
- “other one patients” should be corrected to “the remaining patient”.
- “For all patients, the one-year OS was 54.1% and 2-year OS was 32.4%.”
Please show the graph of OS in this cohort as Figure 1.
- The authors counted one patient who underwent the 3rd RT treatment as two patients in Gender, Child-Pugh score, PVTT, and with/without systemic therapy in Table 1. It is very confusing. The number of patients in this study is 32. In addition, the number of Hepatitis status is 29 and the number of local therapy before 2nd RT is 34. Please correct Table 1. Moreover, the diameter and number of the tumor(s) should be described.
- “For the treatment outcome, 9 lesions had complete remission (CR) and 22 lesions had partial remission (PR) at the image evaluation; these 31 patients were defined as regressive disease (RD) subgroup. One lesion was suggested of progressive disease and the other 5 had no interval change at the image evaluation.”
I speculate that most tumors might be judged as “stable disease” by mRECIST 4-6 weeks after RT. Were the results correct? In addition, “complete response” and “partial response” should be used instead of “complete remission” and “partial remission”.
- “The use of systemic therapy was assumed to benefit on survival…”
I cannot understand why this description suddenly appeared in Results. The authors abovementioned that the use of systemic therapy or not was shown with no benefit on the OS (p=0.307) and IF-PFS (p=0.982).”
Discussion
- Discussion is very confusing, especially in the 1st and 3rd paragraphs. Discussion should be focused on the efficacy and safety of the 2nd-3rd RT therapy for recurrent HCC. In addition, lack of MRI and PET study during the follow-up should be described in the limitation of this study.
- Please compare your results with other treatment modalities.
- “Seol el al.” should be corrected to “Seol et al.”.
Author Response
Comments and Suggestions for Authors
Manuscript review entitled “Hepatic re-irradiation for patients with recurrent hepatocellular carcinoma”
The authors reported the usefulness of 2nd and 3rd RT for recurrent HCCs and the prognostic factors after treatment; however, there are several uncertain points in this manuscript.
General comments
- The details of the cohort is unclear. The authors should clearly describe the patient-tumor backgrounds in this cohort. How many patients did undergo 2nd or 3rd RT for a locally progressed tumor after RT? The diameter of each tumor is also lacking in Table 1.
Thanks for your comment. In the result, line 107 to 111, 32 patients had 2 course of RT and 5 out of these 32 experienced the 3rd course. We’d add to Table 1. We thought for local treatment, volume would be more accurate than diameter in 1 axis out of many slices of image. Thus, we only retrieved the volume.
- I speculate that tumors under several different conditions might be included in this cohort, for example, locally progressed tumors after RT, locally progressed tumors after other treatments (TACE, RFA, and resection), and newly developed tumors at the different site of the liver after RT. The therapeutic effects on abovementioned tumors might be different. The readers want to know whether 2nd RT is effective for the locally progressed tumor after RT. Please describe the details of the efficacy of RT for tumors with different treatment histories.
Thanks for your question. All re-irradiated tumors were recurrent and progressed from previous local treatment. Therefore, it was obvious to state that local treatment before RT failed and it was clear to judge effect of RT. Moreover, these tumors were contradicted against other local therapies, such as surgery, RFA, cryotherapy or TACE; the patients were consulted for RT.
- I speculate that the same patient might be counted as two different patients in Table 1.
Thanks for your question. Five out of 32 patients had the 3rd course, therefore, we took 37 courses into analysis.
- Discussion seems to be very confusing. Please focus on the efficacy and safety of repeated RT. In addition, the overall OS of this study should be compared with other treatments, such as systemic therapy, radioembolization, hepatic infusion chemotherapy, and TACE.
Thanks for your comment and we revised the manuscript. We documented acute toxicity in our result as one patient developed radiation-induced liver disease (RILD) within one month after re-RT in the result.
For development history of local therapy for HCC, RT was after local therapies when contradicting or failure after local therapies. Some novel studies had compared proton therapy or stereotactic ablative radiotherapy to RFA or TACE as the 1st line treatment in more treatment-naïve patients. Our retrospective study recruited more heavily-treated patients (which stated in Table 1) and contradicted against other local therapies, therefore, it would not be our aim and not fair enough to compare with others.
Abstract
- “For treating hepatocellular carcinoma (HCC), liver transplant is the top treatment choice but difficulty of the surgery and availability of liver donation limit the clinical use.”
I do not think so. Surgical resection and ablative therapy are also curative treatment. Please revise the sentence.
Thanks for your advice. We have revised to avoid confuse.
- “Considering systemic therapy and portal vein tumor thrombosis, multivariate analysis only suggested Child-Pugh score and local control were the only two independent factors for the OS with p=0.036 and p=0.022, respectively.”
Why are the words “considering systemic therapy and portal vein tumor thrombus” necessary?
Thanks for your question. Systemic therapy might have benefit for survival and portal vein tumor thrombosis indicates advanced cancer status, we have rephrased the sentence.
Materials and Methods
- “The RT was delivered through IMRT, volumetric modulated arch therapy (VMAT) and tomotherapy, assisted with daily image-guidance.”
The cohort included the patients who were treated with RT in January 2007. Were some early cases treated with conventional RT, not IMRT? Please clarify.
Thanks for your question. All our early patients were treated with IMRT. We have been using IMRT since 2002 and our implement of IGRT won the symbol of National Quality in 2007.
- “Treatment response was evaluated on tri-phase CT scan or MRI 4 to 6 weeks after completion of re-irradiation. Tumor response was determined under the criteria of the modified Response Evaluation Criteria in Solid Tumors (mRECIST) for HCC”
In my experience, tumor vascularity usually remains for a relatively long-term after RT. I think that it is very difficult to judge the local tumor response on CT or MRI obtained 4-6 weeks after RT.
Thanks for your advice. This was also our blind spot which was literature reviewed and discussed in discussion, line 226 – 243.
- Please spell out “RILD”.
Thanks for your reminding.
Results
- “other one patients” should be corrected to “the remaining patient”.
Thanks for your correction.
- “For all patients, the one-year OS was 54.1% and 2-year OS was 32.4%.”
Please show the graph of OS in this cohort as Figure 1.
Thanks for your advice and we have added the figure.
- The authors counted one patient who underwent the 3rd RT treatment as two patients in Gender, Child-Pugh score, PVTT, and with/without systemic therapy in Table 1. It is very confusing. The number of patients in this study is 32. In addition, the number of Hepatitis status is 29 and the number of local therapy before 2nd RT is 34. Please correct Table 1. Moreover, the diameter and number of the tumor(s) should be described.
Thanks for your advice. We used 37 courses as 37 patients for analysis and we’ve specified in table and content. However, some patients experienced OP+TACE, or TACE+RFA before RT. We have revised the table 1.
- “For the treatment outcome, 9 lesions had complete remission (CR) and 22 lesions had partial remission (PR) at the image evaluation; these 31 patients were defined as regressive disease (RD) subgroup. One lesion was suggested of progressive disease and the other 5 had no interval change at the image evaluation.”
I speculate that most tumors might be judged as “stable disease” by mRECIST 4-6 weeks after RT. Were the results correct? In addition, “complete response” and “partial response” should be used instead of “complete remission” and “partial remission”.
Thanks for your advice and we have revised remission to response. Most our evaluation were based on radiologist’s report, though we had discussed this might not be the best response in only 4-6 weeks after treatment.
- “The use of systemic therapy was assumed to benefit on survival…”
I cannot understand why this description suddenly appeared in Results. The authors abovementioned that the use of systemic therapy or not was shown with no benefit on the OS (p=0.307) and IF-PFS (p=0.982).”
Thanks for your comment and we’ve revised the manuscript for avoiding confuse.
Discussion
- Discussion is very confusing, especially in the 1st and 3rd paragraphs. Discussion should be focused on the efficacy and safety of the 2nd-3rd RT therapy for recurrent HCC. In addition, lack of MRI and PET study during the follow-up should be described in the limitation of this study.
Thanks for your comment and we’ve revised the manuscript, mentioning why MRI and PET study were not general used.
- Please compare your results with other treatment modalities.
Thanks for your advice. We mentioned why not comparing with other treatment modalities in question 4. The patients in this study was not suitable for other local therapies; thus it is not objective to compare the result.
- “Seol el al.” should be corrected to “Seol et al.”.
Thanks for your correction.
Reviewer 2 Report
Thank you for your work.
The role of re-irradiation in HCC is a major topic of interest. This retrospective review is of interest and the number of patients is not negligible.
Introduction : The role and place of radiotherapy and re-IR in HCC can be more extensively described.
Material and methods:
Patient characteristics : could authors provide staging information using UICC T stage or BCLC staging as well as number of lesions (number of patients with single vs multiple tumours)
Follow-up and evaluation of toxicity : was there only one Imaging performed at 4-6 weeks with no further follow-up?
Treatment related toxocities : could authors detail the profiles of toxicities being evaluated? do we have data on other common radiation-induced toxicities? (such as fatigue, anorexia, nausea.....) and their grading, those should be presented in a paper evaluating the tolerability and efficacy of re-irradiation
Statistics : Multivariate analysis should include staging information as well as RT doses
Results : Data are missing on dose-response Relationships. Could authors provide data on radiation doses received and tumour response rates according to RT doses or median doses received? It would be of interest to assess impact of radiation dose on OS too.
Use of systemic therapy after 2nd IR : can authors detail systemic therapies received and duration? Based on data presented no significant conclusion can be drawn on the impact of systemic treatment on OS.
Author Response
Thank you for your work.
The role of re-irradiation in HCC is a major topic of interest. This retrospective review is of interest and the number of patients is not negligible.
Introduction:
The role and place of radiotherapy and re-IR in HCC can be more extensively described.
Thanks for your advice and we’ve revised the manuscript.
Material and methods:
Patient characteristics:
could authors provide staging information using UICC T stage or BCLC staging as well as number of lesions (number of patients with single vs multiple tumors)
Thanks for your advice and we’d add the info.
Follow-up and evaluation of toxicity:
was there only one Imaging performed at 4-6 weeks with no further follow-up?
Thanks for your question. There would be image follow-up at the interval of 3 months until progression. We’ve added to the manuscript.
Treatment related toxicities:
could authors detail the profiles of toxicities being evaluated? do we have data on other common radiation-induced toxicities? (such as fatigue, anorexia, nausea...) and their grading, those should be presented in a paper evaluating the tolerability and efficacy of re-irradiation
Thanks for your comment. Classic radiation-induced liver disease has many clinical manifestations, but as a retrospective study, we took a clearer standard and defined as anicteric hepatomegaly and ascites with elevation of alkaline phosphatase, line 95-96. Non-classic RILD was using the definition as elevated serum transaminases (a more than fivefold increase compared to upper limit of normal) and jaundice with elevation of total serum bilirubin, greater 2.5 mg/dL, line 97-98.
Statistics:
Multivariate analysis should include staging information as well as RT doses
Thanks for your advice and we added the data.
Results:
Data are missing on dose-response Relationships. Could authors provide data on radiation doses received and tumor response rates according to RT doses or median doses received? It would be of interest to assess impact of radiation dose on OS too.
Thanks for your advice and the dose data was added. However, due to limited patient number, the analysis suggested dose had no contribution to LC and OS.
Use of systemic therapy after 2nd RT: can authors detail systemic therapies received and duration? Based on data presented no significant conclusion can be drawn on the impact of systemic treatment on OS.
Thanks for your advice and we have added the info of systemic therapy, as well as the duration, in the supplement. However, due to patient number and diversity of drugs, it was difficult to conclude the impact.
Round 2
Reviewer 1 Report
The manuscript has well revised; however, there are several uncertain points in patient background.
Table 1.
- Tumor numbers
The authors described eight patients had two tumors at the 2nd RT. How did the authors simultaneously treat two tumors by RT? Were all two tumors included in a single target volume? Or was each tumor separately treated? Please clarify how to treat two tumors by RT in Materials and Methods.
- BCLC stage
The authors described that 17 patients had BCLC-B HCC. According to Table 1, however, eight patients had two tumors. Did the authors classify a single tumor >5 cm into BCLC-B HCC? If so, please clearly mention in the legend of Table 1. In addition, the grade of portal vein tumor thrombus should be described in Table 1, because it is a significant prognostic factor of HCC.
- Local therapies before the 2nd RT
The authors described that 23 patients had no previous local therapy before the 2nd RT. I recognize that all patients were treated with RT alone before the 2nd RT. Please clarify how many tumors by the 2nd RT were intrahepatic distant recurrence or locally progressed tumors after the 1st RT.
Author Response
The manuscript has well revised; however, there are several uncertain points in patient background.
Table 1.
Tumor numbers
The authors described eight patients had two tumors at the 2nd RT. How did the authors simultaneously treat two tumors by RT? Were all two tumors included in a single target volume? Or was each tumor separately treated? Please clarify how to treat two tumors by RT in Materials and Methods.
Thanks for your question. We’d treat 2 tumors in the same lobe or adjacent segments at the same time, identifying tumor 1 and tumor 2, within one RT treatment plan. Only one patient had 2 tumors at different lobes, which could not treat in the same RT course; this was mentioned in result already. We’ve added more details in Material and Methods. For RT treating 2 tumors, several studies had similar condition. Suggesting one recent study for your reference. (https://www.nature.com/articles/s41598-020-58108-1/tables/1)
BCLC stage
The authors described that 17 patients had BCLC-B HCC. According to Table 1, however, eight patients had two tumors. Did the authors classify a single tumor >5 cm into BCLC-B HCC? If so, please clearly mention in the legend of Table 1. In addition, the grade of portal vein tumor thrombus should be described in Table 1, because it is a significant prognostic factor of HCC.
Thanks for your question. For 8 patients with 2 tumors, 4 of them were BCLC B and the other 4 were BCLC C. All our portal vein tumor thrombosis involved the main portal vein, which would be classified as VP 4 in Japanese grading, at least Type III in Cheng’s classification or at least grade 2 in Yerdel’s grading. Because of the nature of retrospective study, we had difficulty in detailing the extent of PVTT of each patients. We revised the manuscript for better understanding.
Local therapies before the 2nd RT
The authors described that 23 patients had no previous local therapy before the 2nd RT. I recognize that all patients were treated with RT alone before the 2nd RT. Please clarify how many tumors by the 2nd RT were intrahepatic distant recurrence or locally progressed tumors after the 1st RT.
Thanks for your question. Only one patient was re-irradiated on the same progressing tumor while the others were different, intrahepatic recurrent tumors.
Meanwhile, we had some calculation mistakes and uploaded the wrong table for the previous revision; we corrected the mistake and detailed on this revision. We also rephrased as “local therapies before RT” for better understanding.
Reviewer 2 Report
Thank you for the changes and additional data provided.
Additional data on disease classification and systemic treatment is useful and adds value to the study.
In view of the limited number of patients who received systemic treatment (10)and its heterogenity, the conclusion about systemic treatment having no impact on OS and controling distant metastasis should be rephrased including in the abstract as a suggestion rather than a definitive statement. A prospective study and standardized treatment and duration should be used for such definitive conclusions.
Do you have any comments to add in relation to the dose of IR received and response observed in the discussion?
Please also add reference ranges used for Alkaline Phosphatase (ALP).
Some English language rephrasing remains necessary please check.
Author Response
Thank you for the changes and additional data provided.
Additional data on disease classification and systemic treatment is useful and adds value to the study.
Thanks for your advice.
In view of the limited number of patients who received systemic treatment (10)and its heterogenity, the conclusion about systemic treatment having no impact on OS and controling distant metastasis should be rephrased including in the abstract as a suggestion rather than a definitive statement. A prospective study and standardized treatment and duration should be used for such definitive conclusions.
Thanks for your comment and we revised the manuscript.
Do you have any comments to add in relation to the dose of IR received and response observed in the discussion?
Thanks for your remind. Because insignificant finding of dose-response relation in our study and there is no conclusive answer to the question yet, we are reluctant to limit the paragraph in discussing this topic, as well as in consideration to the readers of this journal, with wider background.
Please also add reference ranges used for Alkaline Phosphatase (ALP).
Thanks for your remind. We revised the manuscript.
Some English language rephrasing remains necessary please check.
Thanks for your remind. We rephrased some misunderstanding parts.